# Regulating electron configuration of single Cu sites via unsaturated N,O-coordination for selective oxidation of benzene

Ting Zhang[1], Zhe Sun[1], Shiyan Li[2], Baojun Wang[3,4], Yuefeng Liu [2] ✉, Riguang Zhang [3,4] ✉ & Zhongkui Zhao [1] ✉

Developing highly efficient catalyst for selective oxidation of benzene to phenol (SOBP) with low $H_2O_2$ consumption is highly desirable for practical application, but challenge remains. Herein, we report unique single-atom $Cu_1$-$N_1O_2$ coordination-structure on N/C material (Cu-$N_1O_2$ SA/CN), prepared by water molecule-mediated pre-assembly-pyrolysis method, can efficiently boost SOBP reaction at a 2:1 of low $H_2O_2$/benzene molar ratio, showing 83.7% of high benzene conversion with 98.1% of phenol selectivity. The $Cu_1$-$N_1O_2$ sites can provide a preponderant reaction pathway for SOBP reaction with less steps and lower energy barrier. As a result, it shows an unexpectedly higher turnover frequency (435 $h^{-1}$) than that of $Cu_1$-$N_2$ (190 $h^{-1}$), $Cu_1$-$N_3$ (90 $h^{-1}$) and Cu nanoparticle (58 $h^{-1}$) catalysts, respectively. This work provides a facile and efficient method for regulating the electron configuration of single-atom catalyst and generates a highly active and selective non-precious metal catalyst for industrial production of phenol through selective oxidation of benzene.

Selective oxidation of benzene to phenol (SOBP) with $H_2O_2$ as oxidant is deemed to be a highly efficientand environmental-benign alternative phenol production craft to the conventional cumene process[1-6]. But the general catalytic processes with metallic nanoparticle or complex show unsatisfied results with low activity or phenol selectivity[3-6]. Single-atom catalyst (SAC) is an emerging domain for heterogeneous catalysis[7]. Besides the maximum atom utilization, the SAC can also present catalytic active sites with unique electron structure, thus producing excellent catalytic performance towards diverse reactions[7-11]. Fe, Co, Cu-based SACs have been applied in the SOBP reaction and show considerable catalytic performance[12-18]. However, the results show that the catalytic activity is very low at a low molar ratio of $H_2O_2$/benzene, and a 10:1 or even up to 48:1 of molar ratio of $H_2O_2$/benzene was used to obtain a high benzene conversion[12-18]. If more than 10:1 of $H_2O_2$/benzene molar ratio is used for this reaction, the cost of $H_2O_2$ is far beyond the value of the produced phenol, which leads to this SOBP process not practical application. Therefore, to develop a practical catalyst for SOBP reaction process, the low $H_2O_2$/benzenemolar ratio is essential.

Regulating the electron configuration of single-atom sites through local coordination states adjustment can efficiently modulate the catalytic performance of SACs[19-26]. By changing the coordination number of M-$N_x$ site, researchers found that the coordinatively unsaturated single-atom site features lowered barrier of intermediates formation and products desorption, resulting in improved catalytic performance[19-22]. Besides, heteroatom doping coordination is another efficient strategy for electron configuration regulation[23-26]. For example, with greater electrophilic O coordination, the partially oxidized centralmetal atom possesses more unpaired $d$-electrons which are ready to be excited, resulting in elevated catalytic performance[26]. In our previous work concerning single-atom $Cu_1$-$N_x$ catalyst for SOBP[27,28], the local coordination state of center Cu atom is increased by one Cu-O coordination after reaction, and the recovered catalyst displays stable or improved catalytic performance for SOBP reaction.

[1]State Key Laboratory of Fine Chemicals, Department of Catalysis Chemistry and Engineering, School of Chemical Engineering, Dalian University of Technology, Dalian, PR China. [2]Dalian National Laboratory for Clean Energy (DNL), Dalian Institute of Chemical Physics, Chinese Academy of Science, Dalian, PR China. [3]State Key Laboratory of Clean and Efficient Coal Utilization, Taiyuan University of Technology, Taiyuan, PR China. [4]College of Chemical Engineering and Technology, Taiyuan University of Technology, Taiyuan, PR China. ✉e-mail: yuefeng.liu@dicp.ac.cn; zhangriguang@tyut.edu.cn; zkzhao@dlut.edu.cn

Moreover, a recent reseach theoretically predicts N,O-coordinated Cu single-site is efficient for C-H activation, but lack of practical trials[29]. Therefore, we envision that single Cu sites with unsaturated N,O-coordination might efficiently boost the SOBP reaction at low $H_2O_2$ addition.

Herein, with the purpose of developing an efficient catalyst to boost SOBP reaction at low $H_2O_2$/benzenemolar ratio, we successfully prepare a single-atom Cu catalyst on N/C material with isolated $Cu_1$-$N_1O_2$ sites by a preassembly in aqueous solution followed by a pyrolysis process, confirmed by XAFS, high angle annular dark-field scanning transmission electron microscope (HAADF-STEM), X-ray photoelectron spectroscopy (XPS), and DFT calculation. More interestingly, the as-prepared single-atom catalyst ($Cu$-$N_1O_2$ SA/CN) shows 83.7% of benzene conversion with 98.1% of phenol selectivity at 2:1 of a quite low $H_2O_2$/benzene molar ratio. Furthermore, owing to the unique N,O-coordiantion, the $Cu$-$N_1O_2$ SA/CN catalyst shows 4.8 and 2.3 times higher turnover frequency (*TOF*) value of the previously reported $Cu$-$N_3$ SA/CN and $Cu$-$N_2$ SA/CN catalysts, respectively. We present a practical Cu catalyst for phenol production from selective oxidation of benzene since the excellent catalytic performance can be realized at a quite low $H_2O_2$ addition.

## Results

### Synthesis and structural characterizations

The single-atom $Cu$-$N_1O_2$ SA/CN catalyst was fabricated by a modified preassembly pyrolysis method as early reported[27,28], in which the dimethyl sulfoxide (DMSO) was replaced by deionized water for the supramolecular pre-assembly process, making this procedure environmentally friendly. As shown in Fig. 1a, the melamine aqueous solution with copper nitrate was directly mixed with cyanuric acid aqueous solution, resulting in the Cu containing supermolecule precursor. Owing to the weak basicity of melamine aqueous solution, the $-O^-$ of cyanuric acid molecule was supposed to coordinate with cupric ions, forming Cu-O coordination, besides the coordination of melamine ring with cupric ions. However, in DMSO solvent, the cyanuric acid mole-cule cannot coordinate with cupric ions owing to the very weak coordinating ability of −OH of the cyanuric acid molecule. Fourier Transform Infrared (FTIR) spectroscopy (Supplementary Fig. 1) shows that the C = O stretching bands ($\nu_{C=O}$) located at 1781 and 1741 cm$^{-1}$, which is higher than the reported $\nu_{C=O}$ of cyanuric acid (1739 and 1695 cm$^{-1}$)[30], indicating the formation of hydrogen-bonded supramolecular aggregates via hydrogen bonding of N–H…O and N−H…N linkages between melamine and cyanuric acid[31]. Moreover, the Cu containing $Cu$-$N_1O_2$ SA/CN precursor shows similar FTIR spectrum to that of CN precursor, demonstrating that the presence of $Cu^{2+}$ does not affect the formation of hydrogen-bonded supramolecular aggregates. Followed by pyrolysis under $N_2$ atmosphere at 600 °C for 2 h, the single-atom $Cu$-$N_1O_2$ SA/CN catalyst was obtained. FTIR spectrum of $Cu$-$N_1O_2$ SA/CN (Supplementary Fig. 1) shows that the characteristic peaks of C = O disappear and new peaks centred at 3500–3000, 1800–1100, and 800 cm$^{-1}$ attributed to tri-*s*-triazine arise, proving the presence of a heterocyclic ringstructure[32]. For comparison, bare CN was fabricated without copper nitrate, and supported Cu nanoparticle (Cu NP/CN) catalyst with 0.78 wt% Cu was also fabricated (details see Supplementary)[33].

X-ray diffraction (XRD) patterns (Supplementary Fig. 2) display the characterization diffraction peak at 27.4°, which is related to the stacking of N/C layers, with decreased diffraction intensity for $Cu$-$N_1O_2$ SA/CN contrast to CN and Cu NP/CN, implying the insertion of Cu species into the N/C material (CN) matrix[34–36]. $N_2$-physisorption results (Supplementary Fig. 3 and Supplementary Table 1) show the similar textural properties of as-prepared samples, but larger $V_{BJH}$ value for $Cu$-$N_1O_2$ SA/CN than both CN and Cu NP/CN (0.72 vs 0.35/0.39 cm$^3$ g$^{-1}$), which is attributed to the implanting of Cu species in CN matrix[37]. Scanning electron microscopy (SEM) image (Fig. 1b) demonstrates the similar surface topography of $Cu$-$N_1O_2$ SA/CN to CN and Cu NP/CN with flake and nanotube morphology (Supplementary Fig. 4a, b). Moreover, no nanoparticles or clusters on $Cu$-$N_1O_2$ SA/CN were observed in transmission electron microscopy (TEM) image (Fig. 1c), while Cu nanoparticles on the Cu NP/CN can be clearly detected on

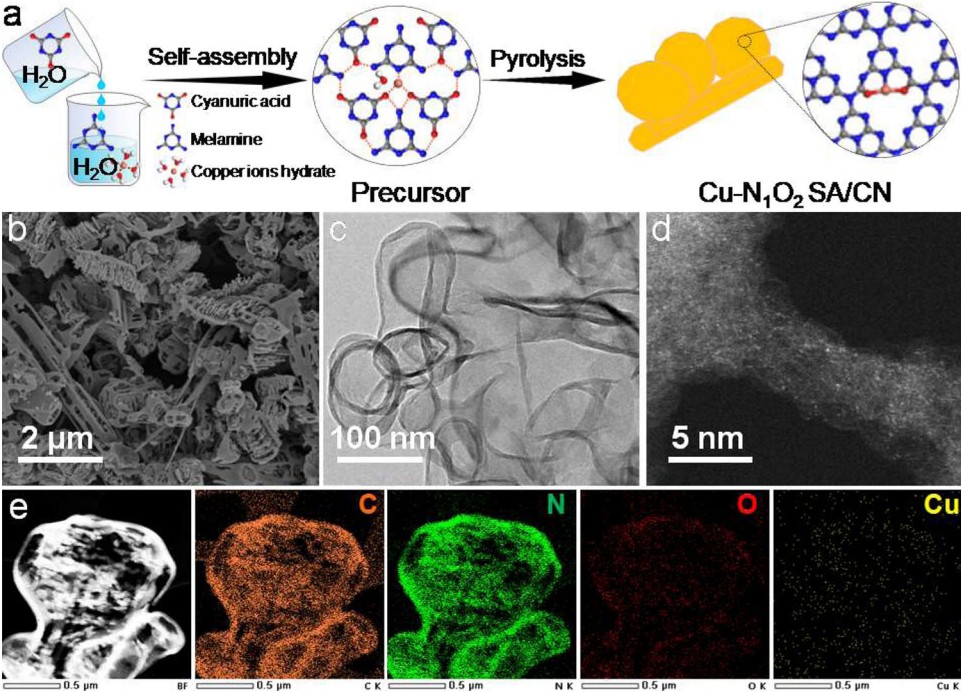

**Fig. 1 | Sample synthesis and morphology characterizations. a** Schematic illustration of preparation of $Cu$-$N_1O_2$ SA/CN. **b** SEM image, **c** HR-TEM image, **d** HAADF-STEM image and **e** the local EDX elemental mappings of $Cu$-$N_1O_2$ SA/CN.

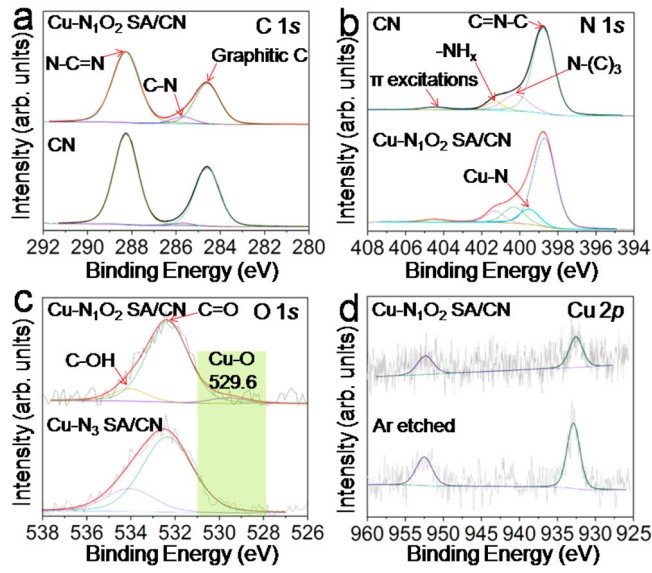

**Fig. 2 | XPS characterizations. a** C 1s and **b** N 1s XPS spectra of Cu-N₁O₂ SA/CN and CN support. **c** O 1s XPS spectra of Cu-N₁O₂ SA/CN and Cu-N₃ SA/CN. **d** Cu 2p XPS spectra of Cu-N₁O₂ SA/CN.

local Energy-Dispersive X-ray (EDX) spectroscopy and TEM images (Supplementary Fig. 4c–e). The Cu atoms are isolated dispersed on CN matrix, which is directly monitored by the atomic-resolution HAADF-STEM image (Fig. 1d, the dense bright dots). Meanwhile, EDX elemental mapping (Fig. 1e) display the uniform C, N, O, Cu distribution on CN matrix of Cu-N₁O₂ SA/CN. The Cu content of Cu-N₁O₂ SA/CN was 0.16 wt%, determined by inductively coupled plasma atomic emission spectroscopy (ICP-AES). KSCN titration (Supplementary Fig. 5 and Supplementary Table 2) was carried out to determine the dispersity of Cu atoms. The results demonstrate that most of Cu atoms are readily accessible.

To explore the electronic properties of catalysts, XPS was conducted and the binding energy (BE) was calibrated by graphitic C at 284.6 eV as internal standard[38,39].The survey XPS spectra (Supplementary Fig. 6) indicate the essential surface elements of C, N, O and similar composition of as-prepared samples (Supplementary Table 3). Deconvoluted C 1s XPS spectra (Fig. 2a) reveal the dominant component of graphitic C (284.6 eV), C-N (285.7 eV) and N-C=N(288.3 eV) species on Cu-N₁O₂ SA/CN, which is consistent with CN and Cu NP/CN (Supplementary Fig. 7a)[40,41]. Deconvoluted N 1s XPS spectrum of Cu-N₁O₂ SA/CN (Fig. 2b) certifies the Cu-N coordination besides C=N-C (398.7 eV), N-(C)₃ (400.3 eV), -NHₓ (401.3 eV) and π excitations (404.5 eV) compared with CN and Cu NP/CN (Supplementary Fig. 8a)[42–44]. Moreover, the deconvoluted O 1s XPS spetrum of Cu-N₁O₂ SA/CN (Fig. 2c) features the Cu-O peak (529.6 eV) compared with Cu-N₃ SA/CN and CN matrix (Supplementary Fig. 9a), indcating the Cu-O coordination. Notably, the Cu-O BE value in Cu-N₁O₂ SA/CN is higher than that in Cu NP/CN (529.6 vs 529.1 eV, Supplementary Fig. 9b), indicating the discrepancy in local coordination environment. The Cu 2p XPS spectra of Cu-N₁O₂ SA/CN (Fig. 2d) and Cu NP/CN (Supplemntary Fig. 8b) show the BE value of ~932.6 eV (between 932.4 eV (Cu⁺ 2p) and 933.6 eV (Cu²⁺ 2p)), implying the low valance state of Cu species (+1<δ < +2)[45,46]. Ar etching treatment on Cu-N₁O₂ SA/CN was carried out (Fig. 2d) and the result shows the increased intensity of Cu 2p signal. Moreover, the Cu 2p signal slightly shift to higher BE value, which might be attributed to the decomposition of CN matrix under Ar etching (Supplementary Fig. 7b).

To further confirm the local coordination structure of atomically dispersed Cu, X-ray absorption near edge structure (XANES) spectra and corresponding fourier transformation of extended X-ray

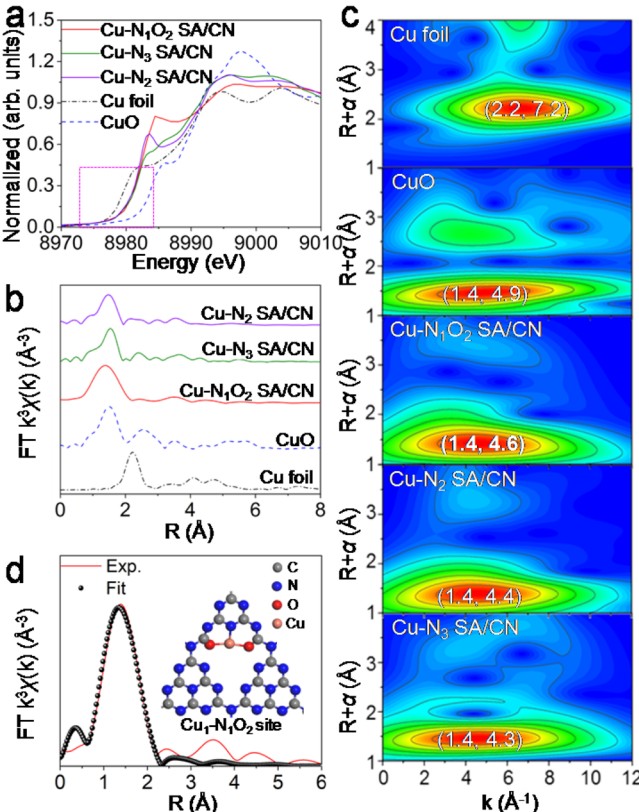

**Fig. 3 | Characterization of local structure and electron properties of single-atom catalyst. a** Normalized Cu K-edge XANES spectra and **b** corresponding k³-weighted Fourier Transform spectra of as-prepared samples. **c** Wavelet transform of Cu-N₁O₂ SA/CN, Cu foil, CuO, Cu-N₂ SA/CN and Cu-N₃ SA/CN. **d** EXAFS fitting curve in R space and the model of Cu₁-N₁O₂ site.

absorption fine structure (FT-EXAFS) spectra were conducted (Fig. 3, Supplementary Figs. 10, 11, Supplementary Table 4). As shown innormalized Cu K-edge XAENS profiles (Fig. 3a), same like the single-atom Cu-N₂ SA/CN and Cu-N₃ SA/CN, the absorption threshold of Cu-N₁O₂ SA/CN located beween Cu foil and CuO (highlighted by dotted), demonstrating the low valance state of Cu[47–50], which is consistent with the XPS result (Fig. 2d). The k³-weighted FT-EXAFS (Fig. 3b) profiles show the single-atom catalysts (Cu-N₂ SA/CN, Cu-N₃ SA/CN and Cu-N₁O₂ SA/CN) feature the main peak at ~1.40 Å, corresponding to the first coordination shell of Cu-N(O). No Cu-Cu (Cu foil) and Cu-O-Cu (CuO) coordination at 2.23 Å and 2.54 Å were observed, demonstrating the atomic Cu on CN matrix. Furthermore, wavelet transform (WT) was performed for the discrimination of backscattering atoms (Fig. 3c and Supplementary Fig. 10c)[51]. Cu-N₁O₂ SA/CN features no Cu-Cu coordination (7.2 Å⁻¹) of Cu foil, further demonstrating the isolated dispersion of Cu atoms. And Cu-N₁O₂ SA/CN displays only one intensity maximum at 4.6 Å⁻¹, which is between the Cu-O (4.9 Å⁻¹) and Cu-N (4.2 Å⁻¹) coordination in CuO and CuPc (Supplementary Fig. 10). In contrast, the N coordinated Cu-N₃ SA/CN and Cu-N₂ SA/CNshow intensity maximun at 4.3 and 4.4 Å⁻¹, respectively. Together with the concomitance of Cu-O and Cu-N bonds demonstrated by XPS results (Fig. 2b, c), the Cu atom was presumed to be coordinated by a mixed structure of Cu-O and Cu-N[52–54]. The quantitative FT-EXAFS fitting analysis (Fig. 3d and Supplementary Table 4) further demonstrates the Cu-N(O) first shell coordination, which is distinctly different from that of Cu foil (Supplementary Fig. 11). The Cu atom was coordinated by 3 neighbouring atoms with the average distance of 1.95 Å. The structure model of single Cu site was constructed by DFT calculations (Supplementary Fig. 12 and Supplementary Table 5), the results show Cu₁-N₁O₂ configuration features lower energy and similar parameters to EXAFS

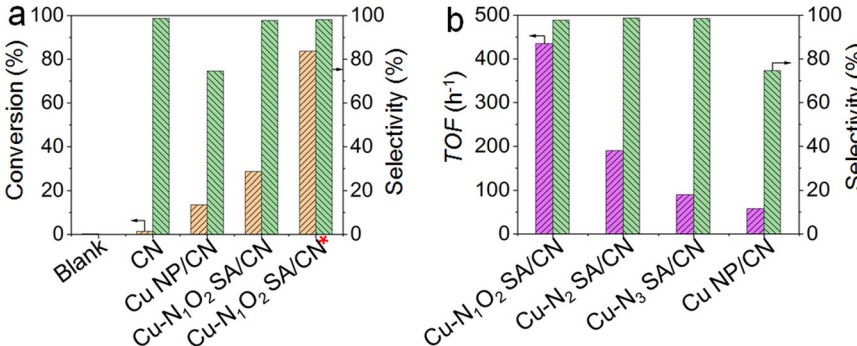

**Fig. 4 | Selective oxidation of benzene to phenol with 2:1 of H₂O₂/benzene molar ratio. a** Catalytic performance over various catalysts for SOBP. Reaction conditions: 30 mg catalyst, 0.4 mL benzene, 2:1 H₂O₂/benzene molar ratio, 2.0 mL CH₃CN as solvent, 60 °C, 5 h. *50 mg catalyst, 72 h, other conditions are same as above. **b** The comparison of benzene TOF over various catalysts. Reaction conditions: 30 mg catalyst, 0.4 mL benzene, 2:1 H₂O₂/benzene molar ratio, 2.0 mL CH₃CN, 60 °C, 5 h.

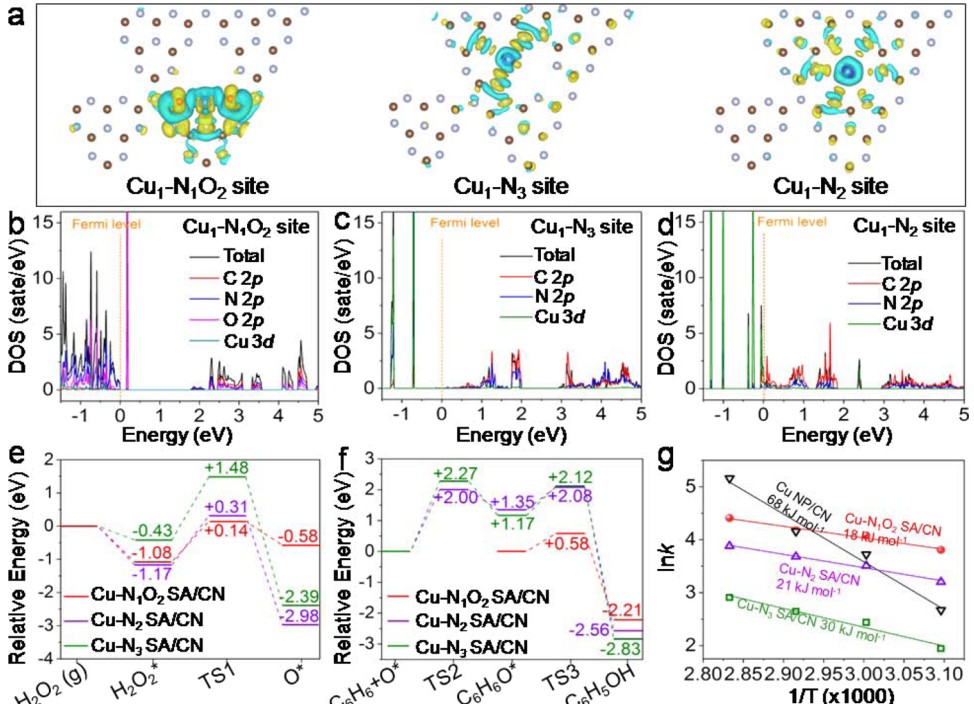

**Fig. 5 | DFT simulations of catalytic activity and electronic structure. a** Differential charge density of Cu₁-N₂, Cu₁-N₃ and Cu₁-N₁O₂ coordination configuration. **b–d** The density of states of Cu₁-N₁O₂ coordination configurations. **e, f** Free energy diagram of H₂O₂ activation **e** and benzene oxidation to phenol **f** on various sites. **g** Arrhenius plots for benzene oxidation over various catalysts. TS: transient state.

fitting results, which is the most possible coordination structure (insert in Fig. 3d).

## Selective oxidation of benzene to phenol

SACs have been proved to be efficient for selective oxidation of benzene to phenol (SOBP), which is deemed to be the atomic economy and clean pathway for phenol production[1–3,12–18]. However, it generally needs large amount of H₂O₂ for high benzene conversion (H₂O₂/Benzene molar ratio > 10) (Supplementary Table 6)[12–18], which makes it not practical in industry. Figure 4a and Supplementary Figs. 13, 14, Supplementary Table 7 display the catalytic performance over various catalysts for SOBP at 2:1 of H₂O₂/Benzene molar ratio. As presented, only trace of benzene is consumed without catalyst (blank) or with CN. The Cu NP/CN shows 13.5% benzene conversion with poor phenol selectivity of 74.6% and carbon balance of 85.4%, indicating its strong

oxidation ability for organics degradation. Interestingly, the developed single-atom Cu-N₁O₂ SA/CN catalyst shows increased benzene conversion of 28.6% with excellent phenol selectivity of 98.0%. It event higher than the single-atom Cu-N₃ SA/CN (19.5%, 98.5%) and Cu-N₂ SA/CN (18.2%, 98.7%). And the benzene conversion gradually increased with the reaction time prolonged, while the phenol selectivity stabilized at ~98% (Supplementary Fig. 14). Moreover, by extending reaction time and increasing catalyst mass, the benzene conversion reached to 83.7% with 98.1% phenol selectivity, demonstrating the superiority of Cu-N₁O₂ SA/CN. Furthermore, the turnover frequency (TOF) results in Fig. 4b show that the Cu-N₁O₂ SA/CN shows 2.3 times TOF of Cu-N₂ SA/CN (435 *vs* 190 h⁻¹) and 4.8 times of Cu-N₃ SA/CN (435 vs 90 h⁻¹), respectively, while the phenol selectivity are similar (~98%). The extrordinary catalytic performance of single-atom Cu-N₁O₂ SA/CN should be attributed to the unique Cu₁-N₁O₂ sites (Supplementary

Tables 8–11 and Supplementary Fig. 15). It can be concluded that the N,O-coordinated singe-atom Cu shows much superior activity to the single-site Cu with Cu-N coordination. Besides the remarkable activity, the Cu-$N_1O_2$ SA/CN also shows high phenol selectivity than Cu NP/CN. Phenol oxidation to benzoquinone over Cu-$N_1O_2$ SA/CN and Cu NP/CN (Supplementary Fig. 16a) demonstrate the weak oxidation ability of Cu-$N_1O_2$ SA/CN for phenol with $H_2O_2$, and the apparent activation energy ($E_a$) of benzene oxidation is lower than that of phenol oxidation over Cu-$N_1O_2$ SA/CN obtained from kinetic study (Supplementary Fig. 16b), which can explain its high selectivity in benzene selective oxidation for phenol. This is the first example for realizing the highly efficient transformation of benzene to phenol through selective oxidation reaction at a 2:1 of low $H_2O_2$/benzene molar ratio, which may promote the industrial production of phenol from benzene selective oxidation.

### Insight into the origin of high catalytic performance

To understand the origin of the remarkable performance over Cu-$N_1O_2$ SA/CN for SOBP, the $H_2O_2$ activation over as-prepared catalysts were eveluated (Supplementary Fig. 17). The single-atom Cu-$N_1O_2$ SA/CN catalyst shows the highest $H_2O_2$ activation ability, which provides large amount of active O* species. Although the Cu NP/CN also shows higher $H_2O_2$ conversion than Cu-$N_2$ SA/CN and Cu-$N_3$ SA/CN, the benzene TOF over Cu NP/CN is the lowest. The severely bubbling in practical operation indicating Cu NP/CN tends to catalyze $H_2O_2$ self-decompostion rather than contributing to hydroxylation. DFT calculations were further performed to study the electronic properties and reaction machenism of various single-atom Cu coordination configurations for the in-depth investigation. Figure 5a displays the differential charge density of Cu$_1$-$N_2$, Cu$_1$-$N_3$ and Cu$_1$-$N_1O_2$ coordination configurations. Owing to the greater electrophilicity of the O atom relative to the N atom, the Cu$_1$-$N_1O_2$ site is more conductive to charge distribution than Cu$_1$-$N_2$ and Cu$_1$-$N_3$ sites, resulting in faster electron transfer between CN support and Cu atoms[55,56]. Furthermore, Bader charge analysis shows that the Cu atom in Cu$_1$-$N_1O_2$ site transfers 0.966 |e| to the neighboring atoms, much higher than Cu$_1$-$N_2$ (0.741 |e|) and Cu$_1$-$N_3$ (0.664 |e|) sites, indicating the better charge transfer capability of Cu$_1$-$N_1O_2$ site. Figure 5b–d display the density of state (DOS) of Cu$_1$-$N_1O_2$, Cu$_1$-$N_3$ and Cu$_1$-$N_2$ coordination configurations. As revealed, the conduction band of Cu$_1$-$N_1O_2$ site is much closer to the Fermi level than that of Cu$_1$-$N_3$ site, further demonstrating the better charge transfer capability of Cu$_1$-$N_1O_2$ site[57]. Moreover, owing to the enhanced charge transfer capability and greater electrophilicity of O atom, the Cu$_1$-$N_1O_2$ site featuring less Cu-3$d$ electron, meaning more 3$d$ orbitals were unoccupied, which is beneficial to the adsorption of reactant[58,59].

Figure 5e, f and Supplementary Fig. 18 display the relative energy profiles of the reaction pathway in the presence of $CH_3CN$ solvent over the Cu$_1$-$N_1O_2$, Cu$_1$-$N_2$ and Cu$_1$-$N_3$ sites. In this study, the solvent effect is considered, and the COSMO (conductor-like solvent model) of Dmol$^3$ is applied to simulate the solvent effects of $CH_3CN$[60,61], and the value of $CH_3CN$ solvent dielectric constant $\varepsilon$ is 37.5 in COSMO. As shown in Fig. 5e, the process of $H_2O_2$ from gaseous state to the adsorbed state is an strongly exothermic over Cu$_1$-$N_1O_2$, Cu$_1$-$N_2$ and Cu$_1$-$N_3$ sites; then, $H_2O_2$* activation to produce O* needs to overcome the activation barrier of 1.15, 0.54, and 1.17 eV over Cu$_1$-$N_1O_2$, Cu$_1$-$N_2$ and Cu$_1$-$N_3$ sites, respectively. However, the strongly exothermic process of $H_2O_2$ from gaseous state to the adsorbed state provide the adequate energy for $H_2O_2$* activation over Cu$_1$-$N_1O_2$ and Cu$_1$-$N_2$ sites, which results in O* production from $H_2O_2$ in gaseous state is a spontaneous process. Over Cu$_1$-$N_3$ site, O* production from $H_2O_2$ in gaseous state requires to overcome the overall barrier of 0.50 eV. In the subsequent benzene oxidation process, the absorbed benzene (absorption energy see Supplementary Table 12) prefer to spontaneously react with active O* species to form $C_6H_6O$* intermediate and one-step produces phenol with an activation barrier of 0.36 eV over Cu$_1$-$N_1O_2$ site (Fig. 5f, phenol

desorption energy see Supplementary Table 12). But for the single Cu$_1$-$N_2$ and Cu$_1$-$N_3$ site, before reacting with the absorbed benzene molecule, the O* species at the stable C-Cu bridge and C-top sites firstly migrate to the Cu center overcoming the higher activation barriers of 1.75 eV and 2.04 eV over single-atom Cu$_1$-$N_2$ and Cu$_1$-$N_3$ site, respectively, then the absorbed benzene molecule is oxidized to phenol overcoming the activation barriers of 0.74 eV and 0.54 eV, respectively (Fig. 5f). Above analysis shows that the overall barrier of benzene oxidation to phenol over Cu$_1$-$N_2$ and Cu$_1$-$N_3$ sites is much higher than that over Cu-$N_1O_2$ site (1.75 and 2.04 eV vs. 0.36 eV), as a result, the single-atom Cu-$N_1O_2$ SA/CN catalyst shows superior catalytic performance than the other twotypesof single-atom samples. Moreover, the $E_a$ over various catalysts obtained from kinetic study (Fig. 5g) demonstrate the lowest $E_a$ value over single-atom Cu-$N_1O_2$ SA/CN catalyst for benzene oxidation.

The good recyclability is important for a heterogeneous catalyst. Supplementary Fig. 19 displays no obvious decrease in benzene conversion over Cu-$N_1O_2$ SA/CN after five cycles, demonstrating its high recycling stability and reusability. XPS and HAADF-STEM concerning the used catalyst (Supplementary Figs. 20 and 21) also reveal that the single Cu atoms remains without agglomeration and the surface properties shows no significant change. And the Cu content in Cu-$N_1O_2$ SA/CN-used shows no obvious erosion (0.14 wt%, determined by ICP-AES).

## Discussion

In summary, we successfully prepared a single-atom Cu catalyst with a unique Cu$_1$-$N_1O_2$ local coordination structure through the preassembly of melamine, cyanuric acid, and copper nitrate in an aqueous solution. The developed catalyst shows 83.7% of benzene conversion with 98.1% of phenol selectivity at 2:1 of a quite low molar ratio of $H_2O_2$/benzene, while more than 10:1 of molar ratio of $H_2O_2$/benzene was generally used to obtain a good reaction result. Owing to the combination of high catalytic performance with the 2:1 of a quite low molar ratio of $H_2O_2$/benzene, this work can boost the practical industrial process for the phenol production through selective oxidation of benzene. DFT calculations reveal the greater electrophilicity of the O atom in Cu$_1$-$N_1O_2$ site endowing single-tom Cu-$N_1O_2$ SA/CN catalyst enhanced charge transfer capability and more unoccupied Cu-3$d$ orbital. As a result, the unique Cu$_1$-$N_1O_2$ moieties provides a preponderant reaction pathway with less steps and lower barrier for SOBP, resulting in the much high TOF value than those on Cu$_1$-$N_2$ and Cu$_1$-$N_3$ sites. We realize highly-efficient benzene-to-phenol transformation at a quite low $H_2O_2$ addition, which promotes the industrial production of phenol through selective oxidation of benzene. For another, this work also opens a new window for designing other single-atom catalysts with unique coordiantion structures towards diverse reactions.

## Methods

### Synthesis of single-atom Cu·N$_1$O$_2$ SA/CN catalyst

The general procedure of fabricating isolated single Cu atoms anchored in N/C material (CN) catalyst was as follow: a certain amount of Cu(NO$_3$)$_2$·3H$_2$O was dissolved in deionized water together with melamine by heating, the obtained solution was marked as solution A. A certain amount of cyanuric acid was dissolved in deionized water by heating, the resulted solution was marked as solution B. Then, solution B was decanted tardily into solution A under stirring condition. The mixture was kept with magnetic stirring. A light green powder precursor was obtained by filtration. The precursor was dried for 12 h after being washed with deionized water and ethanol, respectively. As follows, the as-dried light green powder was acquired. Finally, the powdered precursor was pyrolyzed under N$_2$ atmosphere for 2 h in a tube furnace. The resulted sample was named as Cu-N$_1$O$_2$ SA/CN. The content of Cu is 0.16 wt% determined by ICP-AES.

## Synthesis of single-atom Cu-N₂ SA/CN catalyst

The single-atom Cu-N$_2$ SA/CN was prepared according our former work[28]. Typically, 0.14 g of CN support was dispersed into a copper nitrate containing aqueous sultion in a round-bottom glass flask under magnetic stirring for 40 min. Subsequently, 1 mL of NaBH$_4$ aqueous solution was injected into the flask and kept on stirring for 4 h. After that, solid catalyst was recovered by centrifugation, washed with the deionized water for 3 times and ethanol for 1 time, respectively, and dried at 60 °C overnight. Then, the obtained solid was immersed in 10 mL of dilute HNO$_3$ solution and magnetic stirred for 4 h in a glass flask at room temperature. Then the solid was centrifuged, washed with deionized water to neutral and dried. The finally acquired solid catalyst was named as Cu-N$_2$ SA/CN. The Cu content is 0.20 wt% determined by ICP-AES.

## Synthesis of single-atom Cu-N₃ SA/CN catalyst

The single-atom Cu-N$_3$ SA/CN was prepared according our previous work[27]. Generally, a certain amount of Cu(NO$_3$)$_2$·3H$_2$O was dissolved in 20 mL DMSO with 0.50 g melamine by ultrasonic for 10 min, the as-obtained green clarified solution was marked as solution A. 0.51 g cyanuric acid was dissolved in 10 mL DMSO through ultrasonic for 10 min, the resulted solution was marked as solution B. Then, solution B was decanted tardily into solution A. The green solution momentarily became blue as solution B was added, and white precipitate was formed subsequently. The mixture was kept on magnetic stirring for 10 min. Light green powder precursor was obtained by filtration. The precursor was dried off at 60 °C for 12 h after washed with deionized water and ethanol. Light green dried powder was acquired. Finally, the powdered precursor was pyrolyzed under N$_2$ atmosphere for 4 h in a tube furnace at a ramp rate of 2.3 °C min$^{-1}$. The resulted sample was named as Cu-N$_3$ SA/HCNS. The content of Cu is 0.85 wt% determined by ICP-AES.

## Synthesis of CN support

The procedure of CN synthesis is similar to Cu-N$_1$O$_2$ SA/CN except that without the addition of Cu(NO$_3$)$_2$·3H$_2$O.

## Synthesis of nanoparticle Cu NP/CN catalyst

The Cu NP/CN catalyst was fabricated as former report[27]. Typically, a certain amount of Cu(OAc)$_2$·H$_2$O was dissolved in 18 mL deionized water with 0.43 g PVP-K30 under ultrasonic in a 50 mL round-bottom flask. Then an aqueous solution (2 mL) of 1.2 mmol NaBH$_4$ and 1.0 mmol NaOH was injected into the flask at room temperature and kept stirring for 1 h. Then, 0.5 g CN support was added to the flask and kept on stirring for another 12 h. The solid was centrifuged and washed with water for 3 times and ethanol for 2 times, respectively. The final Cu NP/CN catalyst was obtained after dried off for 12 h at 60 °C. The content of Cu is 0.78 wt% determined by ICP-AES.

**Catalytic performance evaluation.** Selective oxidation of benzene to phenol was performed as the probe evaluation for catalytic performance test. The reaction system includes 30 mg of catalyst, 2 mL of CH$_3$CN, 0.4 mL of benzene and 1 mL of H$_2$O$_2$ (30 wt%). The reaction was carried out at 60 °C in oil bath kettle with magnetic stirring for a period of time. After the reaction was accomplished, the final products were analyzed by Fuli 9790II gas chromatograph (GC) equipped with a 30 m × 0.32 mm × 0.50 μm SE-54 capillary column and a flame ionization detector (FID).

The benzene conversion and phenol selectivity were determined by GC analysis with *n*-dodecane as internal standard.

The conversion of benzene was calculated as: (mole of consumed benzene)/(mole of initial benzene) × 100%.

The selectivity of phenol was calculated as: (mole of formed phenol)/(mole of consumed benzene) × 100%.

The selectivity of benzoquinone was calculated as: (mole of formed benzoquinone)/(mole of consumed benzene) × 100%.

The yield of phenol was calculated as: (mole of formed phenol)/(mole of initial benzene) × 100%.

The carbon balance was calculated as: (mole of formed phenol + mole of formed benzoquinone + mole of remained benzene)/(mole of initial benzene) × 100%.

Turnover frequency (*TOF*) of benzene was calculated as: (mole of consumed benzene)/(reaction time (h) × mole of active Cu).

The mole of active Cu is determined by KSCN titration for single-atom Cu catalyst. For Cu NP/HCNS, the mole of active Cu is 50% for the 2 nm Cu nanoparticles.

## Data availability

All data generated or analyzed in this study are provided in this Article and Supplementary Information, and are also available from the corresponding authors upon request.

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

## Acknowledgements

Z.Z. acknowledges the financial support from the National Natural Science Foundation of China (21978030, 21676046) and the Chinese Ministry of Education via the Program for New Century Excellent Talents in Universities (NCET-12-0079). R.Z. acknowledges the financial support from the National Natural Science Foundation of China (21736007).

## Author contributions

T.Z. conceived and performed the experiments, collected and analyzed data, and wrote the paper. B.W., R.Z., and Z.S. conceived and performed the DFT calculations and wrote the DFT section. S.L. and Y.L. performed the HAADF-STEM measurements. Z.Z. conceived the idea, supervised the project work, and led the data analysis and discussion, and prepared and finalized the paper. All the authors commented on the manuscript and have given approval to the final version of the manuscript.

## Competing interests

The authors declare no competing interests.
