## [Peer Review File · Nature Communications]

Regulating electron configuration of single Cu sites via unsaturated N,O-coordination for selective oxidation of benzeneREVIEWER COMMENTS

Reviewer #1 (Remarks to the Author):

In this work, Zhao et al. prepared a series of Cu single-atom catalysts and studied the influence of the coordination environment of Cu sites on the performance in the oxidation of benzene. Differing in a N-coordination environment, the authors found that the single-atom Cu catalyst with Cu₁-N₁O₂ coordination can efficiently boost the reaction at a low H₂O₂/benzene ratio of 2, and shows a higher turnover frequency than that of the Cu₁-N₂, Cu₁-N₃ and Cu nanoparticle catalysts. The conclusions obtained in the study are interesting; however, this work requires some significant improvement before publication is considered.

1. The Cu 2p XPS signals of Cu-N₁O₂ SA/CN shown in Fig. 2d are too weak to analyze with confidence.
2. EXAFS cannot accurately distinguish N and O atoms. In order to better show the difference in coordination structure and valence state of Cu-N₁O₂ SA/CN, the characterization results of Cu-N₂ SA/CN and Cu-N₃ SA/CN catalysts should also be provided and compared.
3. The content and the type of N species in the four Cu single-atom catalysts are not clarified. Lots of literature have proved that different preparation methods/calcination temperatures have a significant effect on the type and content of the N species. Both of which may lead to a change in catalytic activity.
4. At 1120 cm⁻¹ of the FTIR spectrum, no aromatic ring absorption peak was found as the authors claimed.
5. The specific calculation methods for TON and TOF need to be provided.
6. In this study, KSCN titration was carried out to determine the dispersity of Cu atoms. This method is based on the hypothesis of one Cu site absorbs one SCN⁻¹. Can the author provide relevant literature? How did the author ensure that KSCN was not adsorbed by CN carrier during the test?
7. In addition to the above problems, there are many spelling and stylistic errors. For example, in Lines 103 and 108, "SupplementaryFig.3 and SupplementaryTable1" should be revised to "Supplementary Fig.3 and Supplementary Table1"; In Line 222, "muchcloser" should be revised to "much closer".

Reviewer #2 (Remarks to the Author):

The authors report a highly active and well-characterized SAC for selective benzene oxidation, with TOFs higher than 400 h⁻¹, competitive with benchmark catalysts. The catalytic results are remarkable and the theoretically-driven mechanistic insights offer an understanding of the catalytic steps.

Unfortunately, the use of Cu SACs (on similar heterogeneous supports) for the hydroxylation of benzene into phenol submitted here is already reported in some other studies (some by the authors), both experimentally and theoretically, in line with this work, see for instance: T. Zhang, D. Zhang, X. Han, T. Dong, X. Guo, C. Song, R. Si, W. Liu, Y. Liu, Z. Zhao, *J. Am. Chem. Soc.* 2018, 140, 16936– 16940; T. Zhang, X. Nie, W. Yu, X. Guo, C. Song, R. Si, Y. Liu, Z. Zhao, *iScience* 2019, 22, 97– 108; H. Zhou, Y. Zhao, J. Gan, J. Xu, Y. Wang, H. Lv, S. Fang, Z. Wang, Z. Deng, X. Wang, P. Liu, W. Guo, B. Mao, H. Wang, T. Yao, X. Hong, S. Wei, X. Duan, J. Luo, Y. Wu, *J. Am. Chem. Soc.* 2020, 142, 12643– 12650.

Therefore, given the less urgent and general nature of this nice piece of work, I suggest the publication of the communication in a specialized physical chemistry, catalysis or nanotechnology journal.

Reviewer #3 (Remarks to the Author):

In this study, the authors synthesized a Cu single-atom supported on a N/C material with an unexpected coordination of Cu-N1O2. This catalyst exhibits a high selectivity for benzene oxidation to phenol, a challenging reaction for the chemical industry, with results close to the best catalysts in the literature (83.7% conversion and 98.1% selectivity). More impressive, phenol production is carried out with lower H2O2 amount (2:1 molar ratio) than other reports, which enhances the practical application of this material. The coordination of copper to oxygen atoms is possible due to the acidity of cyanuric acid in water, promoting the coordination via O- sites. The results seem to be relevant for the catalysis field, however, I am not convinced that the benzene conversion and products quantification were properly performed. In order to reach the required level to publish at Nature Communication, it is critical authors address question 5. Thus, the manuscript in the present form is not suitable for publication at Nature Communication. However, considering the relevance of the results, the manuscript might be after addressing the questions below:

1) "As shown in Fig. 1a, the melamine aqueous solution with copper nitrate was directly mixed with cyanuric acid aqueous solution, resulting in the Cu containing supermolecule precursor". The condensation of melamine and cyanuric acid occurs in aqueous media or in the pyrolysis step? What is the difference between light green powder before pyrolysis and the N/C material?

2) "vibration of aromatic ring located 1120 cm⁻¹ of the Cu-N1O2 SA/CN precursor is weaker than that of CN precursor, indicating that Cu species is inlaid in the Cu-N1O2 SA/CN precursor²⁷" The band at 1120 cm⁻¹ is related exactly to which bond? Why the weakening of this band is related to Cu species? The given reference does not include this band in the discussion, not even in the SI. Moreover, it appears from Supplementary Fig.1 that this band does not exist in the Cu-N1O2.

3) "X-ray diffraction (XRD) patterns (Supplementary Fig. 2) display the characterization diffraction peaks of graphitic carbonitride (g-C₃N₄) at 13.0° and 27.4° with decreased diffraction intensity for Cu-N1O2 SA/CN contrast to CN and Cu NP/CN, implying the insertion of Cu species into CN matrix" The peak at 13° is too broad and weak to make any comparison, I would not even consider it as a peak. Also, the authors referred to the material as a carbon nitride, however, the N/C ratio (Supplementary Table 3) is too far from the ideal formula (C₃N₄) and probably the only peak observed (27°) is related to the stacking of N/C layers. The authors must name the material as a N/C material or N-doped carbon rather than a graphitic carbon nitride.

4) "The Cu content of Cu-N1O2 SA/CN was 0.16 wt%, determined by inductively coupled plasma atomic emission spectroscopy (ICP-AES)." The copper loading is quite low, it seems that most of the inserted metal during the synthesis does not coordinate into the CN. What is the yield of copper addition for this synthesis?

5) The authors must provide the conversion equation for the benzene oxidation reaction as well as the other products found in the GC analysis. Does the conversion/selectivity include the possible CO₂ formation? Is the carbon balance closed? The calibration curves used in the quantification must be presented in the SI.

6) In the Supplementary Fig.14, the authors showed that phenol oxidation is inhibited with Cu-N1O2 in comparison to Cu nanoparticles. However, the phenol oxidation conversion is not negligible, since around 10% is oxidized. What are the products found in the phenol oxidation reaction? How the consequent phenol oxidation does not influence the high selectivity of the reaction?

7) The benzene conversion under different reaction times must be provided (only 5h and 72h are shown) to understand the rate of phenol production. Nature Communication publications in the catalysis field are expected to present kinetic data.

8) The authors should carefully revise the text, some sentences in the manuscript contain typos and joined-words, i.e. "The good recyclability is important for a heterogeneous catalyst". Clearly, the text revision was not careful.

Response to the Reviewers' Comments

Reviewer #1 (Remarks to the Author):

In this work, Zhao et al. prepared a series of Cu single-atom catalysts and studied the influence of the coordination environment of Cu sites on the performance in the oxidation of benzene. Differing in a N-coordination environment, the authors found that the single-atom Cu catalyst with Cu₁-N₁O₂ coordination can efficiently boost the reaction at a low H₂O₂/benzene ratio of 2, and shows a higher turnover frequency than that of the Cu₁-N₂, Cu₁-N₃ and Cu nanoparticle catalysts. The conclusions obtained in the study are interesting; however, this work requires some significant improvement before publication is considered.

Response: We appreciate the reviewer's comments. And all the comments and suggestions were considered carefully and incorporated completely in the revision (see below for a point-to-point response).

Comment 1. The Cu 2p XPS signals of Cu-N₁O₂ SA/CN shown in Fig. 2d are too weak to analyze with confidence.

Response: Thanks for your comment. Owing to the low loading of Cu species on Cu-N₁O₂ SA/CN, the corresponding Cu 2p XPS signals are weak even with 10 times of scan in XPS characterization. In the revised version, we applied Ar etching during the XPS operating process and the results were shown in Fig. 2d and Supplementary Fig. 7b.

The Cu 2p XPS spectra of Cu-N₁O₂ SA/CN (Fig. 2d) and Cu NP/CN (supplementary Fig. 8b) show the BE value of ~932.6 eV (between 932.4 eV (Cu⁺ 2p) and 933.6 eV (Cu²⁺ 2p)), implying the low valence state of Cu species (+1< δ <+2) (*Appl. Catal. B Environ.* 2022, **307**, 121154; *ACS Catal.* 2018, **8**, 7113–7119). Ar etching treatment on Cu-N₁O₂ SA/CN was carried out (Fig. 2d) and the result shows the increased intensity of Cu 2p signal. Moreover, the Cu 2p signal slightly shift to higher BE value, which might be attributed to the decomposition of CN matrix under Ar etching (Supplementary Fig. 7b). The changes are highlighted in the revised version by giving the text a yellow background.

Fig. 2 d Cu 2p XPS spectra of Cu-N₁O₂ SA/CN.

Supplementary Fig. 7b The C 1s XPS spectra of Cu-N₁O₂ SA/CN and after Ar etching for 60 s.

Comment 2. EXAFS cannot accurately distinguish N and O atoms. In order to better show the difference in coordination structure and valence state of Cu-N₁O₂ SA/CN, the characterization results of Cu-N₂ SA/CN and Cu-N₃ SA/CN catalysts should also be provided and compared.

Response: Thanks for your constructive comment. The normalized Cu K-edge XANES spectra and corresponding k^3 -weighted FT-EXAFS profiles of single-atom catalysts (Cu-N₂ SA/CN, Cu-N₃ SA/CN and Cu-N₁O₂ SA/CN) were appended in Fig. 3.

As shown in normalized Cu K-edge XANES profiles (Fig. 3a), same like the single-atom Cu-N₂ SA/CN and Cu-N₃ SA/CN, the absorption threshold of Cu-N₁O₂ SA/CN located between Cu foil and CuO (highlighted by dotted), demonstrating the low valence state of Cu (*J. Am. Chem. Soc.* 2017, **139**, 9419–9422; *Angew. Chem. Int. Ed.* 2021, **60**, 1212; *Adv. Funct. Mater.* 2021, **31**, 2100547; *Nat. Commun.* 2021, **12**, 238), which is consistent with the XPS result. The k^3 -weighted FT-EXAFS (Fig. 3b) profiles show the single-atom catalysts (Cu-N₂ SA/CN, Cu-N₃ SA/CN and Cu-N₁O₂ SA/CN) feature the main peak at ~ 1.40 Å, corresponding to the first coordination shell of Cu-N(O). No Cu-Cu (Cu foil) and Cu-O-Cu (CuO) coordination at 2.23 Å and 2.54 Å were observed, demonstrating the atomic Cu on CN matrix. Furthermore, wavelet transform (WT) was performed for the discrimination of backscattering atoms (Fig. 3c and Supplementary Fig. 10c) (*Adv. Funct. Mater.* 2021, **31**, 2100833). Cu-N₁O₂ SA/CN features no Cu-Cu coordination (7.2 Å⁻¹) of Cu foil, further demonstrating the isolated dispersion of Cu atoms. And Cu-N₁O₂ SA/CN displays only one intensity maximum at 4.6 Å⁻¹, which is between the Cu-O (4.9 Å⁻¹) and Cu-N (4.2 Å⁻¹) coordination in CuO and CuPc (Supplementary Fig. 10). In contrast, the N coordinated Cu-N₃ SA/CN and Cu-N₂ SA/CN show intensity maximum at 4.3 and 4.4 Å⁻¹, respectively. Together with the concomitance of Cu-O and Cu-N bonds demonstrated by XPS results (Fig. 2b,c), the Cu atom was presumed to be coordinated by a mixed structure of Cu-O and Cu-N (*Adv. Funct. Mater.* 2021, **32**, 2110224; *Adv. Funct. Mater.* 2022, **32**, 2111446; *ACS Catal.* 2021, **11**, 5212–5221). DFT calculations were further performed and gave the Cu₁-N₁O₂ configuration as the most possible coordination structure (insert in Fig. 3d). The changes are highlighted in the revised version by giving the text a yellow background.

Fig. 3 Characterization of local structure and electron properties of single-atom catalyst. **a** Normalized Cu K-edge XANES spectra and **b** corresponding k^3 -weighted fourier transform spectra of as-prepared samples. **c** Wavelet transform of Cu-N₁O₂ SA/CN, Cu foil, CuO, Cu-N₂ SA/CN and Cu-N₃ SA/CN. **d** EXAFS fitting curve in R space and the model of Cu₁-N₁O₂ site: Cu (Orange), C (Gray), N (Blue), O (Red).

Supplementary Fig. 10 a) Normalized Cu K-edge XANES spectrum, b) corresponding k^3 -weighted Fourier Transform spectrum and c) Wavelet transform of CuPc.

Comment 3. The content and the type of N species in the four Cu single-atom catalysts are not clarified. Lots of literature have proved that different preparation methods/calcination temperatures have a significant effect on the type and content of the N species. Both of which may lead to a change in catalytic activity.

Response: Thanks for your comment. Deconvoluted analysis of N 1s XPS spectra of as-prepared samples (Fig. 2b and supplementary Fig. 8a) feature four types of N species including pyridine N (-C=N-C-), Cu-N, graphitic N (-N-(C)₃) and unsaturated N (-NH_x). The content of different N species in the as-prepared samples has been appended in Supplementary Table 11. As metallic single-atom was generally coordinated with pyridine N species [Adv. Funct. Mater. 2020, 30, 2000768; Chem. Soc. Rev., 2020, 49, 2215-2264; Nat. Commun. 2020, 11, 5283], the pyridine N and Cu-N species were summed up as one. The results in Supplementary Table 11 show slightly distinction among the as-prepared samples, however, no obvious relationship between this distinction and catalytic reactivity (Supplementary Table 7) was revealed. This result further demonstrates that the extraordinary catalytic performance of single-atom Cu-N₁O₂ SA/CN should be attributed to the unique Cu₁-N₁O₂ sites. The changes are highlighted in the revised version by giving the text a yellow background.

Fig. 2b N 1s XPS spectra of Cu-N₁O₂ SA/CN and CN support.

Supplementary Fig. 8a The N 1s XPS spectrum of Cu NP/CN.

Supplementary Table 7. Catalytic performance over various catalysts for SOBP.^[a]

Entry	Sample	Benzene conversion (%)	Selectivity (%)		Phenol yield (%)	Carbon balance (%)
			Phenol	Benzoquinone		
1	Blank	0.2				
2	CN	1.4	98.7	1.1	1.38	99.8
3	Cu NP/CN	13.5	74.6	10.8	10.1	85.4
4	Cu-N ₁ O ₂ SA/CN	28.6	97.8	1.9	28.0	99.7
5	Cu-N ₂ SA/CN	18.2	98.7	0.9	18.0	99.6
6	Cu-N ₃ SA/CN	19.5	98.5	0.8	19.2	99.3
7 ^[b]	Cu-N ₁ O ₂ SA/CN	83.7	98.1	0.9	82.1	99.0

[a] Reaction conditions: 30 mg, 0.4 mL benzene, 2:1 H₂O₂/benzene molar ratio, 2.0 mL CH₃CN as solvent, 60 °C, 5 h.

[b] 50 mg catalyst, 72 h.

Supplementary Table 11. The content of different N species of as-prepared samples.

Entry	Sample	(-C=N-C-) + (Cu-N) (%)	-N-(C) ₃ (%)	-NH _x (%)
1	CN	76	18	6
2	Cu-N ₁ O ₂ SA/CN	80	13	7
3	Cu-N ₂ SA/CN	80	9	11
4	Cu-N ₃ SA/CN	73	22	5

Comment 4. At 1120 cm⁻¹ of the FTIR spectrum, no aromatic ring absorption peak was found as the authors claimed.

Response: Thanks for your comment. We accept the reviewer's comment that the

peak at 1120 cm^{-1} was too weak to be observed even in enlarged view as shown in Fig. R1.

Fig. R1 The enlarge view of FTIR spectra of CN precursor and Cu-N₁O₂ SA/CN precursor.

For accurately describing the results of FTIR, the FTIR spectrum of Cu-N₁O₂ SA/CN catalyst was appended in the Supplementary Fig. 1, as shown followed.

Supplementary Fig. 1 The FTIR spectra of CN precursor, Cu-N₁O₂ SA/CN precursor and Cu-N₁O₂ SA/CN catalyst.

Fourier Transform Infrared (FTIR) spectroscopy (Supplementary Fig. 1) shows that the C=O stretching bands ($\nu_{\text{C=O}}$) located at 1781 and 1741 cm^{-1} , which is higher than the reported $\nu_{\text{C=O}}$ of cyanuric acid (1739 and 1695 cm^{-1}) [*Chem.-Eur. J.* 2009, **15**, 6279–6288], indicating the formation of hydrogen-bonded supramolecular aggregates via hydrogen bonding of N–H...O and N–H...N linkages between melamine and cyanuric acid [*Adv. Funct. Mater.* 2013, **23**, 3661–3667]. Moreover, the Cu containing Cu-N₁O₂ SA/CN precursor shows similar FTIR spectrum to that of CN precursor, demonstrating that the presence of Cu²⁺ does not affect the formation of hydrogen-bonded supramolecular aggregates. Followed by pyrolysis under N₂ atmosphere at $600\text{ }^{\circ}\text{C}$ for 2 h, the single-atom Cu-N₁O₂ SA/CN catalyst was obtained. FTIR spectrum of Cu-N₁O₂ SA/CN (Supplementary Fig. 1) shows that the characteristic peaks of C=O disappear and new peaks centred at $3500\text{--}3000$, $1800\text{--}1100$, and 800 cm^{-1} attributed to tri-s-triazine arise, proving the presence of a heterocyclic ring structure [*J. Am. Chem. Soc.*

2003,125, 10288]. The changes are highlighted in the revised version by giving the text a yellow background.

Comment 5. The specific calculation methods for *TON* and *TOF* need to be provided.

Response: Thanks for your comment. The specific calculation methods for *TON* and *TOF* have been appended in the Methods section and Supporting Information in the revised version and highlighted. Turnover frequency (*TOF*) of benzene was calculated as (mole of consumed benzene)/(reaction time (h)×mole of active Cu).

The turnover number (*TON*) of consumed H₂O₂ was calculated as (mole of consumed H₂O₂)/(mole of active Cu).

The mole of active Cu is determined by KSCN titration for single-atom Cu catalyst. For Cu NP/HCNS, the mole of active Cu is 50% for the 2 nm Cu nanoparticles.

The changes are highlighted in the revised version by giving the text a yellow background.

Comment 6. In this study, KSCN titration was carried out to determine the dispersity of Cu atoms. This method is based on the hypothesis of one Cu site absorbs one SCN⁻¹. Can the author provide relevant literature? How did the author ensure that KSCN was not adsorbed by CN carrier during the test?

Response: Thanks for your constructive comment. The KSCN titration relevant literatures were appended in the revised version [*J. Phys. Chem. Lett.* **2011**, *2*, 295–298; *J. Am. Chem. Soc.* **2017**, *139*, 10790-10798]. These papers demonstrate that the SCN⁻¹ can poison single-atom metal site and catalyst almost deactivated with 1 equivalent KSCN addition. Thus, the hypothesis of one Cu site absorbs one SCN⁻¹ is reasonable.

To eliminate the influence of CN adsorption in KSCN titration, CN carrier was tested and the data was added in Supplementary Table 2. The results indicate minimal effect of CN carrier during the test.

The changes are highlighted in the revised version by giving the text a yellow background.

Supplementary Table 2. Data of KSCN titration for determine the dispersity of Cu on single-atom catalysts

Entry	sample	Mass (mg)	Cu loading (wt%) ^[a]	Abs. (a.u.)	Consumed KSCN (mg)	D _{Cu} (%) ^[b]
1	Cu-N ₃ SA/CN	51.0	0.85	0.696	0.33	49
2	Cu-N ₂ SA/CN	51.4	0.20	0.738	0.14	92
3	Cu-N ₁ O ₂ SA/CN	51.4	0.16	0.748	0.10	79
4	CN	51.8	0	0.768	0.01	-

[a] Determined by ICP-AES. [b] Cu dispersity: (Mole of consumed KSCN) / (Mole of Cu)

Comment 7. In addition to the above problems, there are many spelling and stylistic errors. For example, in Lines 103 and 108, “Supplementary Fig. 3 and Supplementary Table 1” should be revised to “Supplementary Fig. 3 and Supplementary Table1”; In Line 222, "muchcloser" should be revised to "much closer".

Response: Thanks for your constructive comment. The spelling and stylistic errors have been revised and highlighted in the revised version.

Reviewer #2 (Remarks to the Author):

The authors report a highly active and well-characterized SAC for selective benzene oxidation, with *TOFs* higher than 400 h^{-1} , competitive with benchmark catalysts. The catalytic results are remarkable and the theoretically-driven mechanistic insights offer an understanding of the catalytic steps. Unfortunately, the use of Cu SACs (on similar heterogeneous supports) for the hydroxylation of benzene into phenol submitted here is already reported in some other studies (some by the authors), both experimentally and theoretically, in line with this work, see for instance: T. Zhang, D. Zhang, X. Han, T. Dong, X. Guo, C. Song, R. Si, W. Liu, Y. Liu, Z. Zhao, *J. Am. Chem. Soc.* 2018, 140, 16936–16940; T. Zhang, X. Nie, W. Yu, X. Guo, C. Song, R. Si, Y. Liu, Z. Zhao, *iScience* 2019, 22, 97– 108; H. Zhou, Y. Zhao, J. Gan, J. Xu, Y. Wang, H. Lv, S. Fang, Z. Wang, Z. Deng, X. Wang, P. Liu, W. Guo, B. Mao, H. Wang, T. Yao, X. Hong, S. Wei, X. Duan, J. Luo, Y. Wu, *J. Am. Chem. Soc.* 2020, 142, 12643– 12650. Therefore, given the less urgent and general nature of this nice piece of work, I suggest the publication of the communication in a specialized physical chemistry, catalysis or nanotechnology journal.

Response: Thanks for your comment. In this manuscript, we present a highly active and well-characterized single-atom Cu catalyst for selective oxidation of benzene to phenol with high catalytic activity and phenol selectivity at 2:1 of a quite low molar ratio of H_2O_2 /benzene. Moreover, we have in-depth studied the effect of coordination structure of single-atom site on its catalytic performance by theoretical and experimental methods, revealing the importance of electron modulation for designing high efficient catalyst.

Though the use of single-atom Cu catalysts for benzene oxidation have been reported by some other studies, our work shows much progress in both catalytic performance and kinetic understanding. Firstly, we realize highly-efficient benzene-to-phenol transformation at a quite low H_2O_2 addition, which promotes the industrial production of phenol through selective oxidation of benzene. For another, this work also opens a new window for designing other single-atom catalysts with unique coordination structures towards diverse reactions.

Reviewer #3 (Remarks to the Author):

In this study, the authors synthesized a Cu single-atom supported on a N/C material with an unexpected coordination of $\text{Cu-N}_1\text{O}_2$. This catalyst exhibits a high selectivity

for benzene oxidation to phenol, a challenging reaction for the chemical industry, with results close to the best catalysts in the literature (83.7% conversion and 98.1% selectivity). More impressive, phenol production is carried out with lower H_2O_2 amount (2:1 molar ratio) than other reports, which enhances the practical application of this material. The coordination of copper to oxygen atoms is possible due to the acidity of cyanuric acid in water, promoting the coordination via O- sites. The results seem to be relevant for the catalysis field, however, I am not convinced that the benzene conversion and products quantification were properly performed. In order to reach the required level to publish at Nature Communication, it is critical authors address question 5. Thus, the manuscript in the present form is not suitable for publication at Nature Communication. However, considering the relevance of the results, the manuscript might be after addressing the questions below:

Response: We appreciate the reviewer's constructive comments. And all the comments and suggestions were considered carefully and incorporated completely in the revision (see below for a point-to-point response).

Comment 1. "As shown in Fig. 1a, the melamine aqueous solution with copper nitrate was directly mixed with cyanuric acid aqueous solution, resulting in the Cu containing supermolecule precursor". The condensation of melamine and cyanuric acid occurs in aqueous media or in the pyrolysis step? What is the difference between light green powder before pyrolysis and the N/C material?

Response: Thanks for your comment. In the synthesis procedure of $\text{Cu-N}_1\text{O}_2$ SA/CN, the melamine and cyanuric acid were condensed via hydrogen-bond interaction in aqueous media forming supermolecule precursor. With Cu atoms inlaid, it gave a light green color. The difference between the Cu-containing supermolecular precursor and $\text{Cu-N}_1\text{O}_2$ SA/CN catalyst was characterized by FTIR (Supplementary Fig. 1).

Supplementary Fig. 1 The FTIR spectra of CN precursor, $\text{Cu-N}_1\text{O}_2$ SA/CN precursor and $\text{Cu-N}_1\text{O}_2$ SA/CN catalyst.

For the Cu-containing supermolecule precursor ($\text{Cu-N}_1\text{O}_2$ SA/CN precursor), the $\text{C}=\text{O}$ stretching bands ($\nu_{\text{C}=\text{O}}$) at 1781 and 1741 cm^{-1} is higher than the reported $\nu_{\text{C}=\text{O}}$ of cyanuric acid (1739 and 1695 cm^{-1}) [*Chem.-Eur. J.* 2009,**15**, 6279-6288], indicating the formation of hydrogen-bonded supramolecular aggregates via hydrogen bonding of

N–H...O and N–H...N linkages between melamine and cyanuric acid [*Adv. Funct. Mater.*2013, **23**, 3661–3667].

Followed by pyrolysis under N₂ atmosphere at 600 °C for 2 h, the single-atom Cu-N₁O₂ SA/CN catalyst was obtained. During thermal pyrolysis of the supermolecule precursor under protective gas, cyanuric acid reacted with ammonia to give ammelide, ammeline, and finally melamine. Meanwhile, intermediates like melem and melon was formed at 380 °C and 450 °C, respectively. The melon can further condense to yield carbon nitride in the temperature range of 450-600 °C. FTIR spectrum of Cu-N₁O₂ SA/CN shows that the characteristic peaks of C=O disappear and new peaks centred at 3500-3000, 1800-1100, and 800 cm⁻¹ attributed to tri-s-triazine arise, proving the presence of a heterocyclic ring structure [*J. Am. Chem. Soc.*2003, **125**, 10288].

The changes are highlighted in the revised version by giving the text a yellow background.

Comment 2."vibration of aromatic ring located 1120 cm⁻¹ of the Cu-N₁O₂ SA/CN precursor is weaker than that of CN precursor, indicating that Cu species is inlaid in the Cu-N₁O₂ SA/CN precursor²⁷" The band at 1120 cm⁻¹ is related exactly to which bond? Why the weakening of this band is related to Cu species? The given reference does not include this band in the discussion, not even in the SI. Moreover, it appears from Supplementary Fig.1 that this band does not exist in the Cu-N₁O₂.

Response: Thanks for your comment. We accept the reviewer's comment that the peak at 1120 cm⁻¹ was too weak to be observed even in enlarged view as shown in Fig. R1.

Fig. R1 The enlarge view of FTIR spectra of CN precursor and Cu-N₁O₂ SA/CN precursor.

For accurately describing the results of FTIR, the FTIR spectrum of Cu-N₁O₂ SA/CN catalyst was appended in the Supplementary Fig. 1, as shown followed.

Supplementary Fig. 1 The FTIR spectra of CN precursor, Cu-N₁O₂ SA/CN precursor and Cu-N₁O₂ SA/CN catalyst.

Fourier Transform Infrared (FTIR) spectroscopy (Supplementary Fig. 1) shows that the C=O stretching bands ($\nu_{\text{C=O}}$) at 1781 and 1741 cm^{-1} , which is higher than the reported $\nu_{\text{C=O}}$ of cyanuric acid (1739 and 1695 cm^{-1}) [Chem.-Eur. J. 2009, **15**, 6279-6288], indicating the formation of hydrogen-bonded supramolecular aggregates via hydrogen bonding of N-H...O and N-H...N linkages between melamine and cyanuric acid [Adv. Funct. Mater. 2013, **23**, 3661-3667]. Moreover, the Cu containing Cu-N₁O₂ SA/CN precursor shows similar FTIR spectrum to that of CN precursor, demonstrating that the presence of Cu²⁺ does not affect the formation of hydrogen-bonded supramolecular aggregates. Followed by pyrolysis under N₂ atmosphere at 600 °C for 2 h, the single-atom Cu-N₁O₂ SA/CN catalyst was obtained. FTIR spectrum of Cu-N₁O₂ SA/CN (Supplementary Fig. 1) shows that the characteristic peaks of C=O disappear and new peaks centred at 3500-3000, 1800-1100, and 800 cm^{-1} attributed to tri-s-triazine arise, proving the presence of a heterocyclic ring structure [J. Am. Chem. Soc. 2003, **125**, 10288]. The changes are highlighted in the revised version by giving the text a yellow background.

Comment 3. "X-ray diffraction (XRD) patterns (Supplementary Fig. 2) display the characterization diffraction peaks of graphitic carbonitride (g-C₃N₄) at 13.0° and 27.4° with decreased diffraction intensity for Cu-N₁O₂ SA/CN contrast to CN and Cu NP/CN, implying the insertion of Cu species into CN matrix" The peak at 13° is too broad and weak to make any comparison, I would not even consider it as a peak. Also, the authors referred to the material as a carbon nitride, however, the N/C ratio (Supplementary Table 3) is too far from the ideal formula (C₃N₄) and probably the only peak observed (27°) is related to the stacking of N/C layers. The authors must name the material as a N/C material or N-doped carbon rather than a graphitic carbon nitride.

Response: Thanks for your constructive comment. We accept the reviewer's comment that the characterization diffraction peak at 13° in XRD pattern is too broad and weak to be considered as a peak. The peak observed at 27° is related to the stacking of N/C layers (Adv. Funct. Mater. **2013**, *23*, 3661-3667; Appl. Catal. B: Environ. **2023**, *320*, 121928). In the revised version, the matrix was named as N/C material (abbreviated to

CN). The changes are highlighted in the revised version by giving the text a yellow background.

Comment 4. "The Cu content of Cu-N₁O₂ SA/CN was 0.16 wt%, determined by inductively coupled plasma atomic emission spectroscopy (ICP-AES)." The copper loading is quite low, it seems that most of the inserted metal during the synthesis does not coordinate into the CN. What is the yield of copper addition for this synthesis?

Response: Thanks for your comment. In the Cu-N₁O₂ SA/CN synthesis procedure, 40 mg of Cu(NO₃)₂·3H₂O (containing 10.6 mg of Cu) was introduced. After pyrolysis treatment, about 0.15 g of Cu-N₁O₂ SA/CN with Cu loading of 0.16 wt% was obtained, representing 0.24 mg of Cu was inlaid in the final catalyst. Thus, the yield of copper addition in this synthesis is about 2.26%.

The low Cu loading might be due to the follows: In the supramolecular pre-assembly process of melamine with cyanuric acid, there is a competition between the hydrogen-bond interaction of melamine with cyanuric acid and the Cu-O coordination of Cu ions with deprotonated cyanuric acid in alkaline aqueous solution. As a result, only a small number of Cu ions are coordinated and extracted from the Cu nitrate aqueous solution. Therefore, the Cu-N₁O₂ SA/CN has a low Cu content.

The changes are highlighted in the revised version by giving the text a yellow background.

Comment 5. The authors must provide the conversion equation for the benzene oxidation reaction as well as the other products found in the GC analysis. Does the conversion/selectivity include the possible CO₂ formation? Is the carbon balance closed? The calibration curves used in the quantification must be presented in the SI.

Response: Thanks for your constructive comment. The conversion equation for benzene oxidation has been appended in the Methods section and highlighted.

The conversion of benzene was calculated as (mole of consumed benzene)/(mole of initial benzene)×100%.

The selectivity of phenol was calculated as (mole of formed phenol)/(mole of consumed benzene)×100%.

The selectivity of benzoquinone was calculated as (mole of formed benzoquinone)/(mole of consumed benzene)×100%.

The carbon balance was calculated as (mole of formed phenol + mole of formed benzoquinone + mole of remained benzene)/(mole of initial benzene)×100%.

The yield of phenol was calculated as (mole of formed phenol)/(mole of initial benzene)×100%.

The detail data was appended in Supplementary Table 7. As shown, based on our calculation equation, the carbon balance over single-atom catalysts were over 99%, indicating their weak deep oxidation ability for phenol. However, the nanoparticle Cu NP/CN displayed a low carbon balance of 85.4%, demonstrating its strong oxidation ability for organics degradation.

The changes are highlighted in the revised version by giving the text a yellow background.

Supplementary Table 7. Catalytic performance over various catalysts for SOBP.^[a]

Entry	Sample	Benzene conversion (%)	Selectivity (%)		Phenol yield (%)	Carbon balance (%)
			Phenol	Benzoquinone		
1	Blank	0.2				
2	CN	1.4	98.7	1.1	1.38	99.8
3	Cu NP/CN	13.5	74.6	10.8	10.1	85.4
4	Cu-N ₁ O ₂ SA/CN	28.6	97.8	1.9	28.0	99.7
5	Cu-N ₂ SA/CN	18.2	98.7	0.9	18.0	99.6
6	Cu-N ₃ SA/CN	19.5	98.5	0.8	19.2	99.3
7 ^[b]	Cu-N ₁ O ₂ SA/CN	83.7	98.1	0.9	82.1	99.0

[a] Reaction conditions: 30 mg, 0.4 mL benzene, 2:1 H₂O₂/benzene molar ratio, 2.0 mL CH₃CN as solvent, 60 °C, 5 h.

[b] 50 mg catalyst, 72 h.

The calibration curves used in the quantification have been appended in the Supporting Information (See supplementary Fig. 13), as shown followed.

Supplementary Fig. 13 The calibration curves of a) benzene, b) phenol and c) benzoquinone to dodecane for quantification.

Comment 6. In the Supplementary Fig.14, the authors showed that phenol oxidation is inhibited with Cu-N₁O₂ in comparison to Cu nanoparticles. However, the phenol oxidation conversion is not negligible, since around 10% is oxidized. What are the products found in the phenol oxidation reaction? How does the consequent phenol oxidation not influence the high selectivity of the reaction?

Response: Thanks for your comment. The *p*-benzoquinone is the product found in phenol oxidation reaction. For further investigating the origin of high selectivity over Cu-N₁O₂ SA/CN for benzene selective oxidation, the apparent activation energy (E_a) of benzene oxidation and phenol oxidation over Cu-N₁O₂ SA/CN were tested and appended in Supplementary Fig. 16. As revealed, phenol oxidation to benzoquinone over Cu-N₁O₂ SA/CN and Cu NP/CN (Supplementary Fig. 16a) demonstrate the weak oxidation ability of Cu-N₁O₂ SA/CN for phenol with H₂O₂. Moreover, the E_a of benzene oxidation is lower than that of phenol oxidation over Cu-N₁O₂ SA/CN obtained from kinetic study (Supplementary Fig. 16b), which can explain its high selectivity in benzene selective oxidation for phenol. The changes are highlighted in the revised version by giving the text a yellow background.

Supplementary Fig. 16 a) Phenol oxidation with H₂O₂ catalysed with CN, Cu-N₁O₂ SA/CN and Cu NP/CN, respectively. b) Arrhenius plots for benzene oxidation and phenol oxidation over Cu-N₁O₂ SA/CN.

Comment 7. The benzene conversion under different reaction times must be provided (only 5h and 72h are shown) to understand the rate of phenol production. Nature Communication publications in the catalysis field are expected to present kinetic data.

Response: Thanks for your comment. The result of benzene conversion under different reaction times (Supplementary Fig. 14) has been appended in the revised version. As revealed, the benzene conversion gradually increased with the reaction time prolonged, while the phenol selectivity stabilized at ~98%. The changes are highlighted in the revised version by giving the text a yellow background.

Supplementary Fig. 14 The benzene conversion and phenol selectivity over Cu-N₁O₂ SA/CN under different reaction times. Reaction conditions: 30 mg catalyst, 0.4 mL benzene, 2:1 H₂O₂/benzene molar ratio, 2.0 mL CH₃CN as solvent, 60 °C.

Comment 8. The authors should carefully revise the text, some sentences in the manuscript contain typos and joined-words, i.e. "The good recyclability is important for a heterogenous catalyst". Clearly, the text revision was not careful.

Response: Thanks for your constructive comment. The spelling and stylistic errors have been revised and highlighted in the revised version.

REVIEWERS' COMMENTS

Reviewer #1 (Remarks to the Author):

I have carefully gone through the comments and replies. I believe all the issues were fully addressed and the publication is recommended.

Reviewer #3 (Remarks to the Author):

The authors significantly improved the manuscript, most of my concerns were addressed. I recommend its publication.

Response to the Reviewers' Comments

Reviewer #1 (Remarks to the Author):

I have carefully gone through the comments and replies. I believe all the issues were fully addressed and the publication is recommended.

Response: We really appreciate you positive comment on our paper.

Reviewer #3 (Remarks to the Author):

The authors significantly improved the manuscript, most of my concerns were addressed. I recommend its publication.

Response: Thanks for your previous constructive comments to improve our paper and also really appreciate you positive comment on our revised version.